# Vitamin D Status in Pediatric and Young Adult Cystic Fibrosis Patients. Are the New Recommendations Effective?

**DOI:** 10.3390/nu13124413

**Published:** 2021-12-09

**Authors:** Carmen Mangas-Sánchez, María Garriga-García, María Juliana Serrano-Nieto, Ruth García-Romero, Marina Álvarez-Beltrán, Elena Crehuá-Gaudiza, Rosana Muñoz-Codoceo, Lucrecia Suárez-Cortina, Saioa Vicente-Santamaría, Cecilia Martínez-Costa, Juan José Díaz-Martin, Carlos Bousoño-García, David González-Jiménez

**Affiliations:** 1Pediatric Gastroenterology and Nutrition Section, Hospital Central de Asturias, 33011 Oviedo, Spain; carmenmangassanchez@gmail.com (C.M.-S.); ringerbou@yahoo.es (C.B.-G.); domixixon@gmail.com (D.G.-J.); 2Cystic Fibrosis Section, Hospital Ramón y Cajal, 28034 Madrid, Spain; maria.garriga@salud.madrid.org (M.G.-G.); lsuarezcortina@gmail.com (L.S.-C.); saio.vicente@gmail.com (S.V.-S.); 3Pediatric Gastroenterology and Nutrition Section, Hospital Carlos Haya, 29010 Málaga, Spain; serranonieto@hotmail.com; 4Pediatric Gastroenterology and Nutrition Section, Hospital Miguel Servet, 50009 Zaragoza, Spain; ruthgarciaromero@yahoo.es; 5Pediatric Gastroenterology and Nutrition Section, Hospital Vall d’ Hebron, 08035 Barcelona, Spain; marinaalvarezbeltran@gmail.com; 6Pediatric Gastroenterology and Nutrition Section, Hospital Clínico de Valencia, 46010 Valencia, Spain; elenacrehua@gmail.com (E.C.-G.); cecilia.martinez@uv.es (C.M.-C.); 7Pediatric Gastroenterology and Nutrition Section, Hospital Infantil Niño Jesús, 28009 Madrid, Spain; rosana.munozcodoceo@gmail.com

**Keywords:** vitamin D, cystic fibrosis, multicenter study

## Abstract

Introduction: In recent years, guidelines for vitamin D supplementation have been updated and prophylactic recommended doses have been increased in patients with cystic fibrosis (CF). Objective: To evaluate safety and efficacy of these new recommendations. Results: Two cohorts of pancreatic insufficient CF patients were compared before (cohort 1: 179 patients) and after (cohort 2: 71 patients) American CF Foundation and European CF Society recommendations were published. Cohort 2 patients received higher Vitamin D doses: 1509 (1306–1711 95% CI) vs 1084 (983–1184 95% CI) IU/Day (*p* < 0.001), had higher 25 OH vitamin D levels: 30.6 (27.9–33.26 95% CI) vs. 27.4 (25.9–28.8 95% CI) ng/mL (*p* = 0.028), and had a lower prevalence of insufficient vitamin D levels (<30 ng/mL): 48% vs 65% (*p* = 0.011). Adjusted by confounding factors, patients in cohort 1 had a higher risk of vitamin D insufficiency: OR 2.23 (1.09–4.57 95% CI) (*p* = 0.028). Conclusion: After the implementation of new guidelines, CF patients received higher doses of vitamin D and a risk of vitamin D insufficiency decreased. Despite this, almost a third of CF patients still do not reach sufficient serum calcidiol levels.

## 1. Introduction

Vitamin D deficiency is common in CF patients. Between 40% and 90% of these patients have serum 25 OH vitamin D levels below 30 ng/mL [1], while approximately 15–20% have levels below 15 ng/mL [2]. Differences in age groups, type of supplementation, vitamin D3 (cholecalciferol) or vitamin D2 (ergocalciferol), used doses, therapeutic adherence, exocrine pancreatic function and, above all, method for calcidiol measurement and cut-off point used for vitamin D deficiency and insufficiency definition could explain heterogeneity in the observed results.

Patients usually receive supplementation with specific CF polyvitamins that include fat-soluble (vitamin A as β-carotene, vitamin E as α-tocopherol and Vitamin D3) and water soluble vitamins. CF patients often start with Vitamin D supplementation when they present exocrine pancreatic insufficiency, which can be developed over the years, but it normally appears at birth or during the first months of life. For this reason, supplementation usually starts from diagnosis, including those who were diagnosticated by neonatal screening (positive immunoreactive trypsin confirmed by abnormal sweat chloride concentration or identification of a CF disease-causing mutation in each copy of the cystic fibrosis transmembrane conductance regulator (CFTR) gene).

The magnitude of this problem recently led the American CF Foundation and European CF Society to publish two consensus documents on nutritional management of vitamin D in these patients (in 2012 and 2016 respectively) [3,4]. These documents paid particular attention to the following recommendations: First, the importance of supplementation with vitamin D3 instead of vitamin D2 (better results with a greater effect on serum levels and longer lasting effect). Second, monitoring of serum levels should be performed during the winter months. Third, the optimization of exocrine pancreatic function should be achieved. However, one of the main changes was related to the increase in Vitamin D3 dosing for supplementation in these patients, from 400–800 to 400–2000 IU/day depending on age and guideline (Table 1).

The quality of evidence for these new guidelines was poor, since they were based on studies with methodological limitations, carried out mainly in an adult population, and also, in non-CF patients. The Cystic Fibrosis Foundation guideline used the U.S. Preventive Services Task Force grading system (USPSTF) and the degree of certainty for most of the recommendations was low.

The main objective of our study was to evaluate safety and efficacy of these new recommendations regarding vitamin D supplementation in CF pancreatic-insufficient patients.

## 2. Materials and Methods

### 2.1. Study Design

A multicenter retrospective cohort study was conducted in 7 CF Spanish Units. Two cohorts of clinically stable CF pancreatic-insufficient patients were compared before and after the implementation of the new recommendations on vitamin D supplementation: Cohort 1 (according to CFF and ECFS guidelines 2002): years 2012–2013, collected retrospectively from a previous cross-sectional study [5]; Cohort 2 (according to CFF and ECFS guidelines 2012–2016): patients prospectively recruited in the period 2014–2016 for an experimental study. We only included hospital centers that recruited patients for both cohorts.

### 2.2. Patients

#### 2.2.1. Inclusion Criteria

-Patients with a definite CF diagnosis (chloride in sweat >60 mEq/L or positive genetic study) until 40 years old.-Clinical stability defined by clinical criteria: absence of cough, fever, sputum or hemoptysis in the 2 weeks prior to inclusion in the study.-Exocrine pancreatic insufficiency: fecal elastase levels <200 mcg/g.

#### 2.2.2. Exclusion Criteria

-Thrombopenia: platelet count <50,000/mm^3^-Liver dysfunction: elevated liver enzymes >3-fold the upper limit of normality–ULN-; conjugated bilirubin levels >1 mg/dL or liver failure (prothrombin activity <50% or International Normalized Ratio –INR->1.5).-Renal failure (glomerular filtration rate <60 mL/min/1.73 m^2^ or <1 SD of glomerular filtration for age).-Hospital admission or administration of oral or intravenous antibiotics 2 weeks prior to the beginning of the study. -Threatening episode of hemoptysis (any volume of expectorated blood capable of endangering the patients’ life) 4 weeks before the beginning of the study.

### 2.3. Variables

Each patient was identified with a unique code assigned for each hospital and each case. Date of birth, gender, cystic fibrosis transmembrane conductance regulator (CFTR) gene mutation carried, daily dose of vitamin D received, and diagnosis by means of neonatal screening were recorded. 

Weight and height were obtained in the morning with the patient barefoot and in light underwear clothing. Measurement was performed using instruments with a precision of 50 gr and 0.5 cm respectively. Body mass index (BMI) was calculated afterwards. Z-score was obtained for every anthropometric data according to WHO references [6]. The nutritional status of each patient was classified, according to the BMI index in adults and the BMI percentile in children, as: malnourished (<18.5 kg/m^2^; <P10), nutritional risk (18.5–21.9 kg/m^2^; P10–P49), normally nourished (22–24.9 kg/m^2^; P50–P84), overweight (25–29.9 kg/m^2^, P85–P94) and obese (≥30 kg/m^2^; ≥P95) [7,8].

Blood samples were collected after an overnight fast for conventional blood tests (Calcidiol, retinol, alpha-tocopherol, and total cholesterol) and performed at local sites.

Levels of 25 OH vitamin D were determined with a competitive chemiluminescence type immunoassay using a vitamin D binding protein labeled with ruthenium, acridinium or biotin.

Retinol serum levels <30 mcg/dL were classified as vitamin A deficiency [9]. Patients with α-tocopherol / total cholesterol ratio levels below 5.4 mg/g were considered vitamin E deficient [10]. Levels of 25 OH vitamin D less than 30 ng/mL (75 nmol/L) were considered insufficient and less than 20 ng/mL (50 nmol/L) as deficient [11].

Pulmonary function was analyzed using forced spirometry and forced expiratory volume in the first second of expiration (FEV1) was obtained. Percentages were calculated from the absolute values expected for healthy individuals with the same age, gender, and height as the patient. Obstruction was considered when the FEV1 was less than 80% of the theoretic value for the age, height, and weight of the patient [12].

Respiratory samples for conventional bacterial culture were obtained by sputum. In non-collaborating patients, the respiratory sample was obtained by nasopharyngeal swab (<6 years) or sputum induced with hypertonic saline.

### 2.4. Data Analysis:

Data were collected in a Microsoft Office 2010 Access database (Microsoft, Redmond, Washintong, DC, USA) designed ad hoc for the study and were exported to a statistical data management program: STATA version 13.0 (StataCorp LLC, College Station, Texas, TX, USA). Basic statistical techniques for descriptive analysis were applied for the study. Two-tailed t-tests were used for comparison of means and Chi-squared tests were used to compare proportions. To analyse the risk of vitamin D deficiency in cohort 1, a logistic regression analysis was conducted with multivariate adjustment for confounding factors such as age, nutritional status, or infection with *Pseudomonas aeruginosa*. If any of the variables did not meet some of the requirements of normality, non-parametric tests were applied. A *p* value < 0.05 was deemed statistically significant.

## 3. Results

Two groups of patients were included in the study: 179 (cohort 1) vs 71 (cohort 2) (Figure 1).

No significant differences regarding age, sex, genetics, *Pseudomonas aeruginosa* colonization, lung function, or season in which vitamin D was analyzed were observed between groups. A higher proportion of patients diagnosed by neonatal screening was only observed in cohort 2 (54% vs. 27%) (Table 2).

Patients in cohort 2 had higher BMI values and lower nutritional risk than those in cohort 1. However, no differences were observed in the percentage of undernourished (Table 3). Regarding vitamins A and E, no significant differences were observed between groups. Vitamin A serum levels: 39.15 vs 38.44 mcg/dL (*p* = 0.727); vitamin E serum levels: 967.28 vs 952.57 mcg/dL (*p* = 0.798).

Cohort 2 patients received higher Vitamin D doses: 1509 (1306–1711 95% CI) vs 1084 (983–1184 95% CI) IU/Day (*p* < 0.001) and had higher 25 OH vitamin D levels: 30.6 (27.9–33.26 95% CI) vs 27.4 (25.9–28.8 95% CI) (Figure 2a,b).

A higher proportion of patients with insufficient vitamin D serum levels (<30 ng/mL) was observed in cohort 1: 65% vs 48% (*p* = 0.011). The prevalence of patients with vitamin D deficiency (levels <20 ng/mL) was also higher in cohort 1, but these differences were not statistically significant: 25% vs 15% (*p* = 0.1). Adjusting for confounding factors, such as diagnosis by neonatal screening, age, *Pseudomonas aeruginosa* airway colonization, and BMI, patients in cohort 1 had a higher risk of vitamin D insufficiency compared to those in cohort 2 (OR 2.23) (Table 4).

## 4. Discussion

The implementation of new guidelines on Vitamin D supplementation has led to a progressive increase of the vitamin D dose that CF patients receive daily, thus improving the nutritional status of vitamin D in these patients. However, even with these higher doses, almost half of the CF patients do not reach sufficient serum Vitamin D levels (>30 ng/mL).

As expected, according to the new recommendations, patients in cohort 1 received lower doses of vitamin D than those in cohort 2. This increase in dose was associated to an increase in calcidiol levels and a significant improvement in vitamin D status. In group 2, only 48% of the patients showed levels below 30 ng/mL while this prevalence was 65% in group 1.

Our results are consistent with previous studies. Stephenson et al. [13] supplemented 360 adult patients with 400–800 IU/day and only 12% reached levels above 30 ng/mL. Timmers et al. [14] in a study with pediatric patients who received slightly higher Vitamin D doses, from 400–2000 IU/day, 40% of patients had deficient levels (<20 ng/mL), 38.4% insufficient (20–30 ng/mL) and 21.6% sufficient (>30 ng/mL). Brodlie et al. [15] analyzed the impact of increasing the received dose of vitamin D by 475%. An increase in the mean levels of 25 OH vitamin D was observed, but 49% of patients still had insufficient levels, below 30 ng /mL.

Recently, Abu-Fraiha et al. [16] followed a protocol based on CF Foundation recommendations (with initial doses from 800 to 2000 IU / day depending on age; if the 25 Oh vitamin D levels were insufficient they increased the doses between 12 months to 10 years to a maximum of 4000 IU/day, and for those over 10 years old to a maximum of 10,000 IU/day) in 90 exocrine pancreatic-sufficient and insufficient pediatric and adult CF patients. After one year of follow-up, an improvement of mean 25 OH vitamin D levels from 19.5 to 24.5 ng/mL was found. At the end of the study, the percentage of patients with levels below 20 ng/mL decreased from 50% to 27%. Despite the improvement, only 27% of the patients reached levels above 30 ng/mL.

There are multiple factors that determine calcidiol low serum levels in CF patients: poor nutritional intake, decreased outdoor activity, impaired hydroxylation of vitamin D, steroid use, decreased intestinal absorption, or poor compliance to medical treatment [2,17,18,19]. 

In healthy children vitamin D levels vary over the series. Compared with them, CF patients have lower calcidiol levels, especially in children under two years old (healthy children also receive vitamin D supplementation) but in older children the levels become similar [20]. Our results suggested that initial dose is often inadequate for achieving target vitamin D serum levels. According to the updated European Cystic fibrosis recommendation in 2019 [21], it might be convenient to consider a higher initial vitamin D dose (up to 5000 IU/day in >1 year). Other recommendations that may improve nutritional vitamin D status in these patients include optimizing pancreatic enzyme replacement therapy, improving adherence to treatment (in CF, compliance is usually low, especially for fat soluble vitamin supplements. Recent findings show that enhanced therapeutic adherence implies an improvement in vitamin status of CF patients) [22], taking vitamins together with pancreatic enzymes, prescribing cholecalciferol (D3) [23], or changing supplement delivery systems to a water- or powder-based vehicle.

Vitamin D toxicity is defined as calcidiol serum levels above 100–150 ng/mL [3,24]. It is associated with exposure to high doses due to intake or prescription errors [25]. Data about vitamin D toxicity in CF are scarce. A recent study in 244 CF patients found a 5% prevalence of vitamin D toxicity [26]. No patient with vitamin D overdose was observed in our study. All patients included had serum calcidiol levels below 100 ng/mL (up to 95 ng/mL), so our data indicate that upward revision on routine supplementation in these patients, up to 5000 IU/day (highest dose our patients received was 4800 UI/day), is safe.

Vitamin D has anti-inflammatory and antimicrobial properties [27,28]. In CF, some data suggest a clinical relationship between serum calcidiol and pulmonary exacerbations [29,30,31,32] but the association between vitamin D status to lung function is less clear [33,34]. Our data confirm, as in a recent meta-analysis [21], that significant increases of calcidol serum levels after vitamin D treatment in CF did not usually influence the clinical outcomes, e.g., lung function or *Pseudomonas aeruginosa* airway colonization.

The main methodological limitations of our work are marked by the retrospective collection of a series of data that may decrease the accuracy of the same, as well as a probable selection bias due to the different predisposition to participate in the study of the two cohorts that results from recruitment method: cohort 2 patients were recruited prospectively to participate in an experimental study, while those in cohort 1 were recruited retrospectively through the collection of data from medical records. Despite this, it is important to note that the routine follow-up of these patients is very close and is carried out in accredited Cystic Fibrosis units with periodic clinical control every three months. Therefore, the retrospective data are extracted from a reliable source. On the other hand, both cohorts of patients were comparable, without differences in baseline data (except for higher proportion of patients diagnosed by neonatal screening in cohort 2) and regression analyses were performed, including confounding factors. Finally, regarding the probable selection bias, CF patients usually receive supplementation with specific CF polyvitamins that include fat-soluble and water-soluble vitamins, so although therapeutic adherence was not directly analyzed, no differences were observed in serum levels of vitamin A and E between both cohorts, suggesting that adherence was similar. Otherwise, cohort 2 patients should also have higher levels of the rest of fat-soluble vitamins.

## 5. Conclusions

In summary, after the implementation of the new guidelines, higher doses of vitamin D were given to CF patients, which led to a parallel increase in serum levels, and a decreased risk of vitamin D deficiency. Even with this dosing regimen, almost a third of CF patients still show vitamin D insufficiency.

## Figures and Tables

**Figure 1 nutrients-13-04413-f001:**
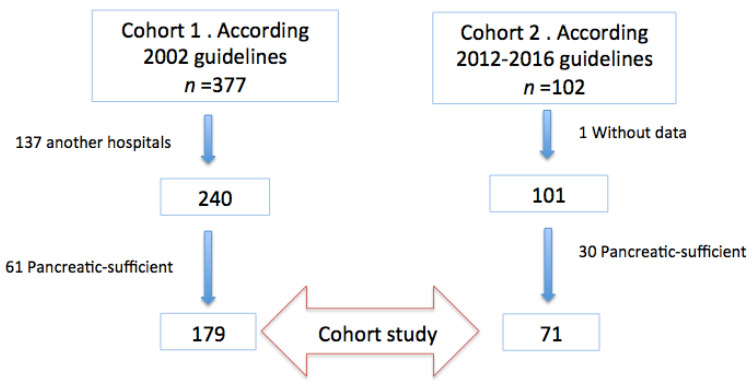
Summary of patient flow diagram.

**Figure 2 nutrients-13-04413-f002:**
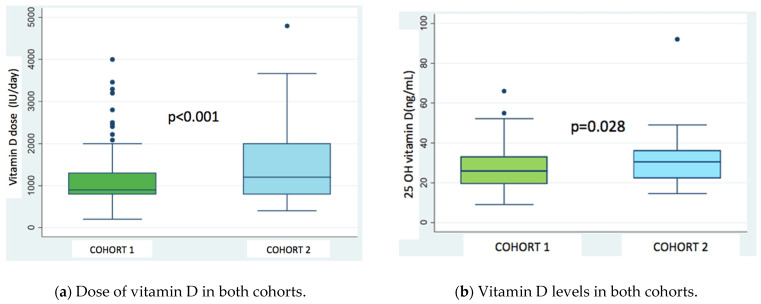
Dose (**a**) and levels (**b**) of vitamin D in both cohorts.

**Table 1 nutrients-13-04413-t001:** Evolution of recommendations on routine supplementation with vitamin D in Cystic Fibrosis patients according to European and American guidelines.

Age	CFF * and ECFS ** Guidelines 2002 (UI/Day)	CFF * Guidelines 2012 (UI/Day)	ECFS ** Guidelines 2016 (UI/Day)
0–12 months	400	400–500	400
1–10 years	400–800	800–1000	800
>10 years	400–800	800–2000	800

* Cystic Fibrosis Foundation; ** European Cystic Fibrosis Society.

**Table 2 nutrients-13-04413-t002:** Main clinical features of both groups.

Variable	Cohort 1 (*n* = 179)	Cohort 2 (*n* = 71)	*p*
Age (year) Mean ± SD (range)	8.6 ± 5.1 (0.1–17.9)	8.3 ± 7.0 (0.1–39.3)	0.703
Age group: % <2 years % 2–10 years % >10 years	14.0 44.7 41.3	19.7 47.9 32.4	0.326
Sex (% female)	45.3	36.6	0.214
Genetic % Homozygous DF508 * % heterozygous DF508 * % Other	38.6 44.7 16.8	43.7 40.9 15.5	0.758
% Neonatal Screening	27.4	54.4	<0.001
Season % Winter % Autumn % Spring % Summer	50.3 14.5 35.2 0	55.0 16.9 26.8 1.4	0.259
*Pseudomonas aeruginosa* airway colonization (%)	15.6	24.2	0.120
FEV **(%) Mean ± SD (range)	90.1 ± 20.7 (35.6–142.0)	85.9 ± 20.9 (41.0–139.0)	0.244
FEV < 80% (%)	29.7	41.9	0.141

* Deletion of phenylalanine 508; ** Forced expiratory volume (FEV).

**Table 3 nutrients-13-04413-t003:** Nutritional status of both cohorts.

Variable	Cohort 1 (*n* = 179)	Cohort 2 (*n* = 71)	*p*
Weight (kg) Mean ± SD (range)	29.25 ± 15.56 (3.1–75)	26.97±16.81 (3.2–74)	0.309
Height (cm) Mean ± SD (range)	125.72 ± 29.5 (50–179.5)	119.094 ± 31.14 (50–177.7)	0.116
BMI (z-score) Mean ± SD (range)	−0.29 ± 0.82 (−2.38–2.58)	0.1 ± 1.02 (−1.9–3.3)	0.004
Cholesterol (mg/dL) Mean ± SD (range)	126 ± 27 (50–230)	125 ± 28 (60–192)	0.781
% Malnourished % Nutritional risk % Normally nourished % Overweight/obese	3.3 70.2 22.1 3.9	4.3 55.3 31.9 8.5	0.741 0.048 0.169 0.203

**Table 4 nutrients-13-04413-t004:** Results of multivariable linear regression for risk of vitamin D insuffiency (serum 25 OH vitamin D < 30 ng/mL).

Variable	Or	95% CI	*p*
Cohort 1	2.23	1.09–4.57	0.028
Age (Years)	0.97	0.91–1.03	0.330
Neonatal screening	1.45	0.73–2.87	0.288
Body mass index (Z-score)	0.94	0.68–1.30	0.712
Pseudomonas aeruginosa airway colonization	1.13	0.52–2.44	0.757
Season: winter	0.84	0.48–1.47	0.539

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
