# Peer review of "Vitamin D Status in Pediatric and Young Adult Cystic Fibrosis Patients. Are the New Recommendations Effective?"

_nutrients, 2021, doi:10.3390/nu13124413_

Round 1
Reviewer 1 Report
The Authors compared the effect of change in Vit D levels following higher Vit D supplementation as per CF guidelines for a cohort of CF patients 2012 to 2-13 and another cohort from 2014-2016
This is an interesting well written retrospective/prospective study.
While cohort are similar in age, gender, CFTR genotype and time of Vit D level.
Major
Authors need to discuss how differences in compliance could have affected different Vit D levels.
Also, I missed a discussion about how the percentage of CF patients with Vit D levels in the insufficient range compare to non-CF children.
Have the authors tried a correlation between Vit D intake and Vit D levels?
Minor
Page 3: line 113: What do you mean with Obstruction was considered when the FEV1 was less than 113 80% of the theoretic value for the age, height and weight of the patient [10].
Figure 1 Exposed and non-exposed is misleading. Can the authors find other descriptors?
Reviewer 2 Report
Abstract:
Line 5, the recommendation was published: What recommendation please clarify.
Introduction:
Please include how CF patients are routinely getting vitamin D supplementation. Is it starting from CF diagnosis or not?
M/M
Line 67-68, exposed vs not exposed: Clarify this, do you mean the two vitamin D regimens? if so, please write it clearly.
Line 69: what is the prospect study? please clarify.
Line 74, without age limit: I thought you were recruiting pediatric and young adults, otherwise, the title may not be appropriate
Variables
Line 91, CFTR: please put the longer version as well.
Line 96, Z-score was obtained for every ... according to WHO reference: please put the reference or the link.
Lines 108-109, levels of vitamin D less than 30ng/ml ... 20ng/ml as deficient (9): Is this a cross reference. The values used in ref 9 is in nmol/l, maybe better to use similar unit.
Results:
Line 136, a higher proportion of patients diagnosed by neonatal screening: Please describe the neonatal screening somewhere probably in the introduction part, including if vitamin D supplementation is started at diagnosis. When were the others diagnosed? This maybe important as it may affect the length of vitamin D supplementation in the 2 cohorts. Please add the length of vitamin D supplementation especially in those not diagnosed by neonatal screening, maybe in table 2.
Table 2: please put the headings in the correct place.
Discussion
Line 182-183, depending on age and vitamin D status, CF foundation recommendation: What did you follow in this study? How were the patients supplemented with vitamin D? Was it according to age and vitamin D status? Please clarify.
Line 206, highest dose in this study was 4800UI/day: Does it mean patients got very different vitamin D doses? Please explain this in the introduction part, how CF patients get vitamin D in routine practice in the study setting.
Line 211: please change in to the
Line 215, different predisposition to participate: What does this mean? Do you mean the method of recruitment, please clarify.
Line 218, routine follow up of these patients is very close: What does it mean? please clarify by describing the routine follow up of the patients recruited in the study.
Line 221, correctly matched: How? please show the matching in the M/M part.
Line 223-224, CF patients receive poly vitamins: please include the details of this in the introduction or M/M part.
